# CERTIFIED ROBUSTNESS OF NEAREST NEIGHBORS AGAINST DATA POISONING ATTACKS

## ABSTRACT

Data poisoning attacks aim to corrupt a machine learning model via modifying, adding, and/or removing some carefully selected training examples, such that the corrupted model predicts any or attacker-chosen incorrect labels for testing examples. The key idea of state-of-the-art certified defenses against data poisoning attacks is to create a *majority vote* mechanism to predict the label of a testing example. Moreover, each voter is a base classifier trained on a subset of the training dataset. Nearest neighbor algorithms such as $k$ nearest neighbors (kNN) and radius nearest neighbors (rNN) have intrinsic majority vote mechanisms. In this work, we show that the intrinsic majority vote mechanisms in kNN and rNN already provide certified robustness guarantees against general data poisoning attacks. Moreover, our empirical evaluation results on MNIST and CIFAR10 show that the intrinsic certified robustness guarantees of kNN and rNN outperform those provided by state-of-the-art certified defenses.

## 1 INTRODUCTION

Data poisoning attacks (Barreno et al., 2006; Nelson et al., 2008; Biggio et al., 2012; 2013a; Xiao et al., 2015b; Steinhardt et al., 2017; Shafahi et al., 2018) aim to corrupt the training phase of a machine learning system via carefully poisoning its training dataset including modifying, adding, and/or removing some training examples. The corrupted model predicts incorrect labels for testing examples. Data poisoning attacks pose severe security concerns to machine learning in critical application domains such as autonomous driving (Gu et al., 2017), cybersecurity (Rubinstein et al., 2009; Suciu et al., 2018; Chen et al., 2017), and healthcare analytics (Mozaffari-Kermani et al., 2014). Unlike adversarial examples (Szegedy et al., 2014; Goodfellow et al., 2014; Carlini & Wagner, 2017), which add perturbation to each testing example to induce misclassification, data poisoning attacks corrupt the model such that it misclassifies many clean testing examples.

Multiple certifiably robust learning algorithms (Ma et al., 2019; Rosenfeld et al., 2020; Levine & Feizi, 2020; Jia et al., 2020) against data poisoning attacks were recently developed. A learning algorithm is certifiably robust against data poisoning attacks if it can learn a classifier on a training dataset that achieves a *certified accuracy* on a testing dataset when the number of poisoned training examples is no more than a threshold (called *poisoning size*). The certified accuracy of a learning algorithm is a lower bound of the accuracy of its learnt classifier no matter how an attacker poisons the training examples with the given poisoning size. The key idea of state-of-the-art certifiably robust learning algorithms (Levine & Feizi, 2020; Jia et al., 2020) is to create a *majority vote* mechanism to predict the label of a testing example. In particular, each voter votes a label for a testing example and the final predicted label is the majority vote among multiple voters. For instance, Deep Partition Aggregation (DPA) (Levine & Feizi, 2020) divides the training dataset into disjoint partitions and learns a base classifier (i.e., a voter) on each partition. Bagging (Jia et al., 2020) also learns multiple base classifiers (i.e., voters), but each of them is learnt on a random subsample of the training dataset. We denote by $a$ and $b$ the labels with the largest and second largest number of votes, respectively. Moreover, $s_a$ and $s_b$ respectively are the number of votes for labels $a$ and $b$ when there are no corrupted voters. The corrupted voters change their votes from $a$ to $b$ in the worst-case scenario. Therefore, the majority vote result (i.e., the predicted label for a testing example) remains to be $a$ when the number of corrupted voters is no larger than $\lceil \frac{s_a - s_b}{2} \rceil - 1$. In other words, the number of corrupted voters that a majority vote mechanism can tolerate depends on the gap $s_a - s_b$ between the largest and the second largest number of votes.

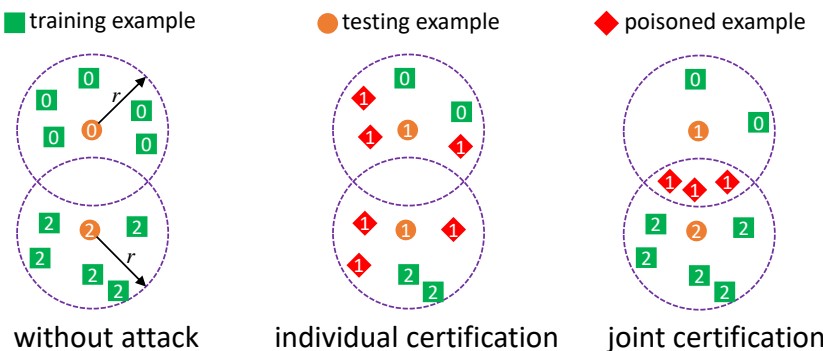

**Figure 1: An example to illustrate individual certification vs. joint certification. Suppose rNN correctly classifies the two testing examples without attack. An attacker can poison 3 training examples. The attacker can make rNN misclassify each testing example individually. However, the attacker cannot make rNN misclassify both testing examples jointly.**

However, state-of-the-art certifiably robust learning algorithms achieve suboptimal certified accuracies due to two key limitations. First, *each poisoned training example leads to multiple corrupted voters* in the worse-case scenarios. In particular, modifying a training example corrupts two voters (i.e., two base classifiers) in DPA (Levine & Feizi, 2020) and corrupts the voters whose training subsamples include the modified training example in bagging (Jia et al., 2020). Therefore, given the same gap $s_a - s_b$ between the largest and the second largest number of votes, the majority vote result is robust against a small number of poisoned training examples. Second, *they can only certify robustness for each testing example individually* because it is hard to quantify how poisoned training examples corrupt the voters for different testing examples jointly. Suppose the classifier learnt by a learning algorithm can correctly classify testing inputs $\mathbf{x}_1$ and $\mathbf{x}_2$. An attacker can poison $e$ training examples such that the learnt classifier misclassifies $\mathbf{x}_1$ or $\mathbf{x}_2$, but the attacker cannot poison $e$ training examples such that both $\mathbf{x}_1$ and $\mathbf{x}_2$ are misclassified. When the poisoning size is $e$, existing certifiably robust learning algorithms would produce a certified accuracy of 0 for the two testing examples. However, the certified accuracy can be 1/2 if we consider them jointly.

$k$ nearest neighbors (kNN) and radius nearest neighbors (rNN) (Fix & Hodges, 1951; Cover & Hart, 1967) have intrinsic majority vote mechanisms. Specifically, given a testing example, kNN (or rNN) predicts its label via taking a majority vote among the labels of its $k$ nearest neighbors (or neighbors within radius $r$) in the training dataset. Our major contribution in this work is that we show the intrinsic majority vote mechanisms in kNN and rNN make them certifiably robust against data poisoning attacks. Moreover, kNN and rNN address the limitations of state-of-the-art certifiably robust learning algorithms. Specifically, each poisoned training example leads to only one corrupted voter in the worse-case scenario in kNN and rNN. Therefore, given the same gap $s_a - s_b$, the majority vote result (i.e., predicted label for a testing example) is robust against more poisoned training examples in kNN and rNN. Furthermore, we show that rNN enables joint certification of multiple testing examples. Figure 1 illustrates an example of individual certification and joint certification with two testing examples in rNN. When we treat the two testing examples individually, an attacker can poison 3 training examples such that rNN misclassifies each of them. However, when we treat them jointly, an attacker cannot poison 3 training examples to misclassify both of them. We propose such joint certification to derive a better certified accuracy for rNN. Specifically, we design methods to group testing examples in a testing dataset such that we can perform joint certification for each group of testing examples.

We evaluate our methods on MNIST and CIFAR10 datasets. We use $\ell_1$ distance metric to calculate nearest neighbors. First, our methods substantially outperform state-of-the-art certifiably robust learning algorithms. For instance, when an attacker can arbitrarily poison 1,000 training examples on MNIST, the certified accuracy of rNN with $r = 4$ is 40.8% and 33.5% higher than those of DPA (Levine & Feizi, 2020) and bagging (Jia et al., 2020), respectively. Second, our joint certification improves certified accuracy. For instance, our joint certification improves the certified accuracy of rNN by 15.1% when an attacker can arbitrarily poison 1,000 training examples on MNIST.

In summary, we make the following contributions:

- We derive the intrinsic certified robustness guarantees of kNN and rNN against data poisoning attacks.

- We propose joint certification of multiple testing examples to derive a better certified robustness guarantee for rNN. rNN is the first method that supports joint certification of multiple testing examples against data poisoning attacks.

- We evaluate our methods and compare them with state-of-the-art on MNIST and CIFAR10.

## 2 PROBLEM SETUP

**Learning setting:** Assuming we have a training dataset $D_{tr}$ with $n$ training examples. We denote by $\mathcal{M}$ a learning algorithm. Moreover, we denote by $\mathcal{M}(D_{tr}, \mathbf{x})$ the label predicted for a testing input $\mathbf{x}$ by a classifier learnt by $\mathcal{M}$ on the training dataset $D_{tr}$. For instance, given a training dataset $D_{tr}$ and a testing input $\mathbf{x}$, kNN finds the $k$ training examples in $D_{tr}$ that are the closest to $\mathbf{x}$ as the nearest neighbors, while rNN finds the training examples in $D_{tr}$ whose distances to $\mathbf{x}$ are no larger than $r$ as the nearest neighbors. The distance between a training input and a testing input can be measured by any distance metric. Then, kNN and rNN use majority vote among the nearest neighbors to predict the label of $\mathbf{x}$. Specifically, each nearest neighbor is a voter and votes its label for the testing input $\mathbf{x}$; and the label with the largest number of votes is the final predicted label for $\mathbf{x}$.

**Data poisoning attacks:** We consider data poisoning attacks (Rubinstein et al., 2009; Biggio et al., 2012; Xiao et al., 2015a; Li et al., 2016; Muñoz-González et al., 2017; Jagielski et al., 2018) that aim to poison (i.e., modify, add, and/or remove) some carefully selected training examples in $D_{tr}$ such that the corrupted classifier has a low accuracy for testing examples indiscriminately. For simplicity, we use $D_{tr}^*$ to denote the *poisoned training dataset*. Moreover, we define the *poisoning size $e$* as the minimal number of modified/added/removed training examples that can turn $D_{tr}$ into $D_{tr}^*$. We use $S(D_{tr}, e)$ to denote the set of poisoned training datasets whose poisoning sizes are at most $e$. Formally, we define $S(D_{tr}, e)$ as follows:

$$S(D_{tr}, e) = \{D_{tr}^* | \max\{|D_{tr}^*|, |D_{tr}|\} - |D_{tr}^* \cap D_{tr}| \le e\}, \tag{1}$$

where $\max\{|D_{tr}^*|, |D_{tr}|\} - |D_{tr}^* \cap D_{tr}|$ is the poisoning size of $D_{tr}^*$. Note that modifying a training example is equivalent to removing a training example and adding a new one. Given a training dataset $D_{tr}$ and a poisoning size $e$, an attacker aims to craft a poisoned training dataset $D_{tr}^*$ to minimize the testing accuracy of the classifier learnt by algorithm $\mathcal{M}$ on $D_{tr}^*$.

**Certified accuracy:** Given a training dataset $D_{tr}$ and a learning algorithm $\mathcal{M}$, we use *certified accuracy* on a testing dataset $D_{te} = \{(\mathbf{x}_i, y_i)\}_{i=1}^t$ to measure the algorithm's performance. Specifically, we denote *certified accuracy at poisoning size $e$* as $CA(e)$ and formally define it as follows:

$$CA(e) = \min_{D_{tr}^* \in S(D_{tr}, e)} \frac{\sum_{(\mathbf{x}_i, y_i) \in D_{te}} \mathbb{I}(\mathcal{M}(D_{tr}^*, \mathbf{x}_i) = y_i)}{|D_{te}|}, \tag{2}$$

where $\mathbb{I}$ is the indicator function and $\mathcal{M}(D_{tr}^*, \mathbf{x}_i)$ is the label predicted for a testing input $\mathbf{x}_i$ by the classifier learnt by the algorithm $\mathcal{M}$ on the poisoned training dataset $D_{tr}^*$. $CA(e)$ is the least testing accuracy on $D_{te}$ that the learning algorithm $\mathcal{M}$ can achieve no matter how an attacker poisons the training examples when the poisoning size is at most $e$. Our goal is to derive lower bounds of $CA(e)$ for learning algorithms kNN and rNN.

## 3 CERTIFIED ACCURACY OF KNN AND RNN

We first derive a lower bound of the certified accuracy via *individual certification*, which treats testing examples in $D_{te}$ individually. Then, we derive a better lower bound of the certified accuracy for rNN via *joint certification*, which treats testing examples jointly.

### 3.1 INDIVIDUAL CERTIFICATION

Given a poisoning size at most $e$, our idea is to certify whether the predicted label stays unchanged or not for each testing input individually. If the predicted label of a testing input $\mathbf{x}$ stays unchanged

(i.e., $\mathcal{M}(D_{tr}, \mathbf{x}) = \mathcal{M}(D_{tr}^*, \mathbf{x})$) and it matches with the testing input's true label, then kNN or rNN certifiably correctly classifies the testing input when the poisoning size is at most $e$. Therefore, we can obtain a lower bound of the certified accuracy at poisoning size $e$ as the fraction of testing inputs in $D_{te}$ which kNN or rNN certifiably correctly classifies. Next, we first discuss how to certify whether the predicted label stays unchanged or not for each testing input individually. Then, we show our lower bound of the certified accuracy at poisoning size $e$.

**Certifying the predicted label of a testing input:** Our goal is to certify that $\mathcal{M}(D_{tr}, \mathbf{x}) = \mathcal{M}(D_{tr}^*, \mathbf{x})$ for a testing input $\mathbf{x}$ when the poisoning size is no larger than a threshold. Given a training dataset $D_{tr}$ (or a poisoned training dataset $D_{tr}^*$) and a testing input $\mathbf{x}$, we use $\mathcal{N}(D_{tr}, \mathbf{x})$ (or $\mathcal{N}(D_{tr}^*, \mathbf{x})$) to denote the set of nearest neighbors of $\mathbf{x}$ in $D_{tr}$ (or $D_{tr}^*$) for kNN or rNN. We note that there may exist ties when determining the nearest neighbors for kNN, i.e., multiple training examples may have the same distance to the testing input. Usually, kNN breaks such ties uniformly at random. However, such random ties breaking method introduces randomness, i.e., the difference of nearest neighbors before and after poisoned training examples (i.e., $\mathcal{N}(D_{tr}, \mathbf{x})$ vs. $\mathcal{N}(D_{tr}^*, \mathbf{x})$) depends on the randomness in breaking ties. Such randomness makes it challenging to certify the robustness of the predicted label against poisoned training examples. To address the challenge, we propose to define a deterministic ranking of training examples and break ties via choosing the training examples with larger ranks. Moreover, such ranking between clean training examples does not depend on poisoned ones. For instance, we can use a cryptographic hash function (e.g., SHA-1) that is very unlikely to have collisions to hash each training example based on its input feature vector and label, and then we rank the training examples based on their hash values.

We use $s_l$ to denote the number of votes in $\mathcal{N}(D_{tr}, \mathbf{x})$ for label $l$, i.e., the number of nearest neighbors in $\mathcal{N}(D_{tr}, \mathbf{x})$ whose labels are $l$. Formally, we have $s_l = \sum_{(\mathbf{x}_j, y_j) \in \mathcal{N}(D_{tr}, \mathbf{x})} \mathbb{I}(y_j = l)$, where $l = 1, 2, \cdots, c$ and $\mathbb{I}$ is an indicator function. kNN or rNN essentially predicts the label of the testing input $\mathbf{x}$ as the label with the largest number of votes, i.e., $\mathcal{M}(D_{tr}, \mathbf{x}) = \arg\max_{l \in \{1,2,\cdots,c\}} s_l$. Suppose $a$ and $b$ are the labels with the largest and second largest number of votes, i.e., $s_a$ and $s_b$ are the largest and second largest ones among $\{s_1, s_2, \cdots, s_c\}$, respectively. We note that there may exist ties when comparing the labels based on their votes. We define a deterministic ranking of labels in $\{1, 2, \cdots, c\}$ and take the label with the largest rank when such ties happen. In the worse-case scenario, each poisoned training example leads to one corrupted voter in kNN or rNN, which changes its vote from label $a$ to label $b$. Therefore, kNN or rNN still predicts label $a$ for the testing input $\mathbf{x}$ when the number of poisoned training examples is no more than $\lceil \frac{s_a - s_b}{2} \rceil - 1$ (without considering the ties breaking). Formally, we have the following theorem:

**Theorem 1.** *Assuming we have a training dataset $D_{tr}$, a testing input $\mathbf{x}$, and a nearest neighbor algorithm $\mathcal{M}$ (i.e., kNN or rNN). $a$ and $b$ respectively are the two labels with the largest and second largest number of votes among the nearest neighbors $\mathcal{N}(D_{tr}, \mathbf{x})$ of $\mathbf{x}$ in $D_{tr}$. Moreover, $s_a$ and $s_b$ are the number of votes for $a$ and $b$, respectively. Then, we have the following:*

$$\mathcal{M}(D_{tr}^*, \mathbf{x}) = a, \forall D_{tr}^* \in S(D_{tr}, e), e \leq \lceil \frac{s_a - s_b + \mathbb{I}(a > b)}{2} \rceil - 1. \tag{3}$$

*Proof.* When an attacker poisons at most $e$ training examples, the number of changed nearest neighbors in $\mathcal{N}(D_{tr}, \mathbf{x})$ is at most $e$. We denote by $s_l^* = \sum_{(\mathbf{x}_j, y_j) \in \mathcal{N}(D_{tr}^*, \mathbf{x})} \mathbb{I}(l = y_j)$ the number of votes for label $l$ among the nearest neighbors $\mathcal{N}(D_{tr}^*, \mathbf{x})$ in the poisoned training dataset, where $l = 1, 2, \cdots, c$. Then, we have $s_l - e \leq s_l^* \leq s_l + e$ for each $l = 1, 2, \cdots, c$. Therefore, when $e \leq \lceil \frac{s_a - s_b + \mathbb{I}(a > b)}{2} \rceil - 1$, we have $s_a^* - s_b^* \geq s_a - s_b - 2 \cdot e > 0$ if $a < b$ and $s_a^* - s_b^* \geq s_a - s_b - 2 \cdot e \geq 0$ if $a > b$. Thus, the nearest neighbor algorithm still predicts label $a$ for $\mathbf{x}$ in both cases based on our way of breaking ties, i.e., we have $\mathcal{M}(D_{tr}^*, \mathbf{x}) = a$ when $e \leq \lceil \frac{s_a - s_b + \mathbb{I}(a > b)}{2} \rceil - 1$. $\qquad\square$

**Deriving a lower bound of $CA(e)$:** kNN or rNN certifiably correctly classifies a testing input $\mathbf{x}$ if it correctly predicts its label before attacks and the predicted label stays unchanged after an attacker poisons the training dataset. Therefore, the fraction of testing inputs that kNN or rNN certifiably correctly classifies is a lower bound of $CA(e)$. Formally, we have the following theorem:

**Theorem 2** (Individual Certification). *Assuming we have a training dataset $D_{tr}$, a testing dataset $D_{te} = \{(\mathbf{x}_i, y_i)\}_{i=1}^t$, and a nearest neighbor algorithm $\mathcal{M}$ (i.e., kNN or rNN). $a_i$ and $b_i$ respectively are the two labels with the largest and second largest number of votes among the nearest neighbors*

$\mathcal{N}(D_{tr}, \mathbf{x}_i)$ *of* $\mathbf{x}_i$ *in* $D_{tr}$. *Moreover,* $s_{a_i}$ *and* $s_{b_i}$ *are the number of votes for* $a_i$ *and* $b_i$, *respectively. Then, we have the following lower bound of* $CA(e)$:

$$CA(e) \geq \frac{\sum_{(\mathbf{x}_i, y_i) \in D_{te}} \mathbb{I}(a_i = y_i) \cdot \mathbb{I}(e \leq e_i^*)}{|D_{te}|}, \tag{4}$$

*where* $e_i^* = \lceil \frac{s_{a_i} - s_{b_i} + \mathbb{I}(a_i > b_i)}{2} \rceil - 1$.

*Proof.* See Appendix A. □

### 3.2 JOINT CERTIFICATION

We derive a better lower bound of the certified accuracy via jointly considering multiple testing examples. Our intuition is that, given a group of testing examples and a poisoning size $e$, an attacker may not be able to make a learning algorithm misclassify all the testing examples jointly even if it can make the learning algorithm misclassify each of them individually. In particular, rNN enables such joint certification. It is challenging to perform joint certification for kNN because of the complex interactions between the nearest neighbors of different testing examples (see our proof of Theorem 3 for specific reasons). Next, we first derive a lower bound of $CA(e)$ on a group of testing examples for rNN. Then, we derive a lower bound of $CA(e)$ on the testing dataset $D_{te}$ via dividing it into groups. Finally, we discuss different strategies to divide the testing dataset into groups, which may lead to different lower bounds of $CA(e)$.

**Deriving a lower bound of** $CA(e)$ **for a group of testing examples:** Suppose we have a group of testing examples which have different predicted labels in rNN. Our key intuition is that when an attacker can poison $e$ training examples, the attacker can only decrease the total votes for the testing examples' predicted labels by at most $e$ in rNN, as the testing examples' predicted labels are different. We denote by $\mathcal{U}$ a group of testing examples with different predicted labels and by $m$ its size, i.e., $m = |\mathcal{U}|$. The next theorem shows a lower bound of $CA(e)$ on the testing examples in $\mathcal{U}$ for rNN.

**Theorem 3.** *Assuming we have a training dataset* $D_{tr}$, *the learning algorithm rNN, and a group of* $m$ *testing examples* $\mathcal{U} = \{(\mathbf{x}_i, y_i)\}_{i=1}^m$ *with different predicted labels.* $a_i$ *and* $b_i$ *respectively are the two labels with the largest and second largest number of votes among the nearest neighbors* $\mathcal{N}(D_{tr}, \mathbf{x}_i)$ *of* $\mathbf{x}_i$ *in* $D_{tr}$. *Moreover,* $s_{a_i}$ *and* $s_{b_i}$ *are the number of votes for* $a_i$ *and* $b_i$, *respectively. Without loss of generality, we assume the following:*

$$(s_{a_1} - s_{b_1}) \cdot \mathbb{I}(a_1 = y_1) \geq (s_{a_2} - s_{b_2}) \cdot \mathbb{I}(a_2 = y_2) \geq \cdots \geq (s_{a_m} - s_{b_m}) \cdot \mathbb{I}(a_m = y_m). \tag{5}$$

*Then, the certified accuracy at poisoning size* $e$ *of rNN for* $\mathcal{U}$ *has a lower bound* $CA(e) \geq \frac{w-1}{|\mathcal{U}|}$, *where* $w$ *is the solution to the following optimization problem:*

$$w = \underset{w', w' \geq 1}{\arg\min} w' \quad s.t. \quad \sum_{i=w'}^m \max(s_{a_i} - s_{b_i} - e + \mathbb{I}(a_i > b_i), 0) \cdot \mathbb{I}(a_i = y_i) \leq e. \tag{6}$$

*Proof.* When an attacker can poison at most $e$ training examples, the attacker can add at most $e$ new nearest neighbors and remove $e$ existing ones in $\mathcal{N}(D_{tr}, \mathbf{x}_i)$ (equivalent to modifying $e$ training examples) in the worst-case scenario. We denote by $s_{a_i}^*$ and $s_{b_i}^*$ respectively the number of votes for labels $a_i$ and $b_i$ among the nearest neighbors $\mathcal{N}(D_{tr}^*, \mathbf{x}_i)$. First, we have $s_{b_i}^* \leq s_{b_i} + e$ for $\forall i \in \{1, 2, \cdots, m\}$ since at most $e$ new nearest neighbors are added. Second, we have $s_{a_i}^* \geq s_{a_i} - e_i$ in rNN, where $e_i$ is the number of removed nearest neighbors in $\mathcal{N}(D_{tr}, \mathbf{x}_i)$ whose true labels are $a_i$. Note that kNN does not support joint certification because $s_{a_i}^* \geq s_{a_i} - e_i$ does not hold for kNN.

Next, we derive the minimal value of $e_i$ such that rNN misclassifies $\mathbf{x}_i$. In particular, we consider two cases. If $a_i \neq y_i$, i.e., $\mathbf{x}_i$ is misclassified by rNN without attack, then we have $e_i = 0$. If $a_i = y_i$, $\mathbf{x}_i$ is misclassified by rNN when $s_{a_i}^* \leq s_{b_i}^*$ if $a_i < b_i$ and $s_{a_i}^* < s_{b_i}^*$ if $a_i > b_i$ after attack, which means $e_i \geq s_{a_i} - s_{b_i} - e + \mathbb{I}(a_i > b_i)$. Since $e_i \geq 0$, we have $e_i \geq \max(s_{a_i} - s_{b_i} - e + \mathbb{I}(a_i > b_i), 0)$. Combining the two cases, we have the following lower bound for $e_i$ that makes rNN misclassify $\mathbf{x}_i$: $e_i \geq \max(s_{a_i} - s_{b_i} - e + \mathbb{I}(a_i > b_i), 0) \cdot \mathbb{I}(a_i = y_i)$. Moreover, since the attacker can remove at most $e$ training examples and the group of testing examples have different predicted labels, i.e., $a_i \neq a_j \; \forall i, j \in \{1, 2, \cdots, m\}$ and $i \neq j$, we have $\sum_{i=1}^m e_i \leq e$. We note that the lower bound of

$e_i$ is non-increasing as $i$ increases based on Equation (5). Therefore, in the worst-case scenario, the attacker can make rNN misclassify the last $m - w + 1$ testing inputs whose corresponding $e_i$ sum to be at most $e$. Formally, $w$ is the solution to the optimization problem in Equation (6). Therefore, the certified accuracy at poisoning size $e$ is at least $\frac{w-1}{|\mathcal{U}|}$. $\qquad\qquad\square$

**Deriving a lower bound of $CA(e)$ for a testing dataset:** Based on Theorem 3, we can derive a lower bound of $CA(e)$ for a testing dataset via dividing it into disjoint groups, each of which includes testing examples with different predicted labels in rNN. Formally, we have the following theorem:

**Theorem 4** (Joint Certification). *Given a testing dataset $D_{te}$, we divide it into $\lambda$ disjoint groups, i.e., $\mathcal{U}_1, \mathcal{U}_2, \cdots, \mathcal{U}_\lambda$, where the testing examples in each group have different predicted labels in rNN. Then, we have the following lower bound of $CA(e)$:*

$$CA(e) \geq \frac{\sum_{j=1}^{\lambda} \mu_j \cdot |\mathcal{U}_j|}{\sum_{j=1}^{\lambda} |\mathcal{U}_j|}, \tag{7}$$

*where $\mu_j$ is the lower bound of the certified accuracy at poisoning size $e$ on group $\mathcal{U}_j$, which we can obtain by invoking Theorem 3.*

*Proof.* See Appendix B. $\qquad\qquad\square$

**Strategies of grouping testing examples:** Our Theorem 4 is applicable to any way of dividing the testing examples in $D_{te}$ to disjoint groups once the testing examples in each group have different predicted labels in rNN. Therefore, a natural question is how to group the testing examples in $D_{te}$ to maximize our lower bound of certified accuracy. For instance, a naive method is to randomly divide the testing examples into disjoint groups, each of which includes at most $c$ (the number of classes) testing examples with different predicted labels. We call such method *Random Division (RD)*. However, RD achieves suboptimal performance because it does not consider the certified robustness of each individual testing example. In particular, some testing examples can or cannot be certifiably correctly classified no matter which groups they belong to. However, if we group them with other testing examples, the certified accuracy may be degraded because each group can have at most $c$ testing examples. For instance, if a testing example cannot be certifiably correctly classified no matter which group it belongs to, then adding it to a group would exclude another testing example from the group, which may degrade the certified accuracy for the group.

Therefore, we propose to isolate these testing examples and divide the remaining testing examples into disjoint groups. We call such method *Isolation and Division (ISLAND)*. Specifically, we first divide the testing dataset $D_{te}$ into three disjoint parts which we denote as $D_{te}^0$, $D_{te}^1$, and $D_{te}^2$. $D_{te}^0$ contains the testing examples that cannot be certifiably correctly classified at poisoning size $e$ no matter which group they belong to. Based on our proof of Theorem 3, a testing example $(\mathbf{x}_i, y_i)$ that satisfies $(s_{a_i} - s_{b_i} - e + \mathbb{I}(a_i > b_i)) \cdot \mathbb{I}(a_i = y_i) \leq 0$ cannot be certifiably correctly classified at poisoning size $e$ no matter which group it belongs to. Therefore, $D_{te}^0$ includes such testing examples. Moreover, based on Theorem 1, a testing example $(\mathbf{x}_i, y_i)$ that satisfies $e \leq \lceil \frac{s_{a_i} - s_{b_i} + \mathbb{I}(a_i > b_i)}{2} \rceil - 1$ can be certifiably correctly classified at poisoning size $e$. Therefore, $D_{te}^1$ includes such testing examples. Each testing example in $D_{te}^0$ or $D_{te}^1$ forms a group by itself. $D_{te}^2$ includes the remaining testing examples, which we further divide into groups. Our method of dividing $D_{te}^2$ into groups is inspired by the proof of Theorem 3. In particular, we form a group of testing examples as follows: for each label $l \in \{1, 2, \cdots, c\}$, we find the testing example that has the largest value of $(s_{a_i} - s_{b_i} - e + \mathbb{I}(a_i > b_i)) \cdot \mathbb{I}(a_i = l)$ and we skip the label if there is no remaining testing example whose predicted label is $l$. We apply the procedure to recursively group the testing examples in $D_{te}^2$ until no testing examples are left.

## 4 EVALUATION

**Datasets:** We evaluate our methods on MNIST and CIFAR10. We use the popular histogram of oriented gradients (HOG)[1] (Dalal & Triggs, 2005) method to extract features for each example, which

---

[1]Public implementation: https://scikit-image.org/docs/dev/api/skimage.feature.html#skimage.feature.hog

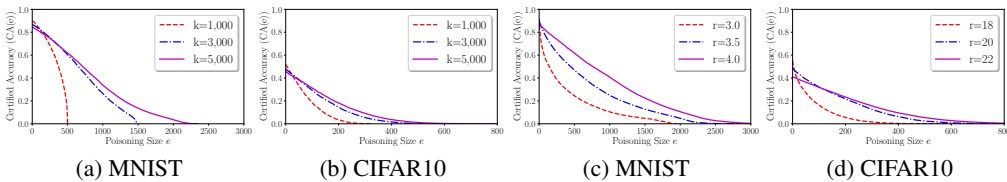

(a) MNIST     (b) CIFAR10     (c) MNIST     (d) CIFAR10

**Figure 2: (a)–(b) comparing kNN and rNN with state-of-the-art methods. (c)–(d) comparing individual certification with joint certification for rNN.**

(a) MNIST     (b) CIFAR10     (c) MNIST     (d) CIFAR10

**Figure 3: Impact of $k$ ((a)–(b)) and $r$ ((c)–(d)) on the certified accuracy of kNN and rNN.**

we found improves certified accuracy. Note that previous work (Jia et al., 2020) used a pre-trained model to extract features via transfer learning. However, the pre-trained model may also be poisoned and thus we don't use it. We didn't find ties in determining nearest neighbors for kNN in our experiments. We rank the labels as $\{1, 2, \cdots, 10\}$ to break ties for labels.

**Parameter settings:** While any distance metric is applicable, we use $\ell_1$ in our experiments for both kNN and rNN. Unless otherwise mentioned, we adopt the following settings: $k = 5,000$ for both MNIST and CIFAR10 in kNN; and $r = 4$ for MNIST and $r = 20$ for CIFAR10 in rNN, considering the different feature dimensions of MNIST and CIFAR10. By default, we use the ISLAND grouping method in the joint certification for rNN.

**Comparing with DPA (Levine & Feizi, 2020) and bagging (Jia et al., 2020):** Figure 2a and Figure 2b show the comparison results of DPA, bagging, kNN, and rNN. DPA divides a training dataset into $\zeta$ disjoint partitions and learns a base classifier on each of them. Then, DPA takes a majority vote among the base classifiers to predict the label of a testing example. Bagging learns $N$ base classifiers, each of which is learnt on a random subsample with $\xi$ training examples of the training dataset. Moreover, bagging's certified accuracy is correct with a confidence level $1 - \alpha$. All the compared methods have tradeoffs between accuracy under no attacks (i.e., $CA(0)$) and robustness against attacks. Therefore, we set their parameters such that they have similar accuracy under no attacks (i.e., similar $CA(0)$). In particular, we use the default $k$ for kNN, and we adjust $r$ for rNN, $\zeta$ for DPA, and $\xi$ for bagging. The searched parameters are as follows: $r = 4$, $\zeta = 5,500$, and $\xi = 27$ for MNIST; and $r = 21$, $\zeta = 900$, and $\xi = 400$ for CIFAR10. Note that we set $N = 1,000$ and $\alpha = 0.001$ for bagging following (Jia et al., 2020).

We have the following observations. First, both kNN and rNN outperform DPA and bagging. The superior performance of kNN and rNN stems from two reasons: 1) each poisoned training example corrupts multiple voters for DPA and bagging, while it only corrupts one voter for kNN and rNN, which means that, given the same gap between the largest and second largest number of votes, kNN and rNN can tolerate more poisoned training examples; and 2) rNN enables joint certification that improves the certified accuracy. Second, rNN achieves better certified accuracy than kNN when the poisoning size is large. The reason is that rNN supports joint certification.

**Comparing individual certification with joint certification:** Figure 2c and Figure 2d compare individual certification and joint certification (with the RD and ISLAND grouping methods) for rNN. Our empirical results validate that joint certification improves the certified accuracy upon individual certification. Moreover, our ISLAND grouping method outperforms the RD method.

**Impact of $k$ and $r$:** Figure 3 shows the impact of $k$ and $r$ on the certified accuracy of kNN and rNN, respectively. As the results show, $k$ and $r$ achieve tradeoffs between accuracy under no attacks (i.e., $CA(0)$) and robustness. Specifically, when $k$ or $r$ is smaller, the accuracy under no attacks, i.e., $CA(0)$, is larger, but the certified accuracy decreases more quickly as the poisoning size $e$ increases.

## 5 RELATED WORK

Data poisoning attacks have been proposed against various learning algorithms such as Bayes classifier (Nelson et al., 2008), SVM (Biggio et al., 2012), clustering (Biggio et al., 2013b; 2014), collaborative filtering (Li et al., 2016), regression models (Xiao et al., 2015a; Mei & Zhu, 2015b; Jagielski et al., 2018), LDA (Mei & Zhu, 2015a), neural networks (Muñoz-González et al., 2017; Shafahi et al., 2018; Suciu et al., 2018; Demontis et al., 2019; Zhu et al., 2019; Huang et al., 2020), and others (Rubinstein et al., 2009; Vuurens et al., 2011).

To mitigate data poisoning attacks, many empirical defenses (Cretu et al., 2008; Rubinstein et al., 2009; Barreno et al., 2010; Biggio et al., 2011; Feng et al., 2014; Jagielski et al., 2018; Tran et al., 2018) have been proposed. Steinhardt et al. (2017) derived an upper bound of the loss function for data poisoning attacks when the model is learnt using examples in a feasible set. However, these defenses lack certified robustness guarantees. Recently, several certified defenses (Ma et al., 2019; Rosenfeld et al., 2020; Levine & Feizi, 2020; Jia et al., 2020) were proposed to defend against data poisoning attacks. These defenses provide certified accuracies for a testing dataset either probabilistically (Ma et al., 2019; Jia et al., 2020) or deterministically (Rosenfeld et al., 2020; Levine & Feizi, 2020). All these defenses except (Ma et al., 2019) create majority vote mechanisms to predict the label of a testing example. In particular, a voter is a base classifier learnt on a perturbed version of the training dataset in randomized smoothing based defenses (Rosenfeld et al., 2020), while a voter is a base classifier learnt on a subset of the trainig dataset in DPA (Levine & Feizi, 2020) and bagging (Jia et al., 2020). Ma et al. (2019) showed that a differentially private learning algorithm achieves certified accuracy against data poisoning attacks. They also train multiple differentially private classifiers, but they are not used to predict the label of a testing example via majority vote. Instead, their average accuracy is used to estimate the certified accuracy. kNN and rNN have intrinsic majority vote mechanisms and we show that they provide deterministic certified accuracies against data poisoning attacks. Moreover, rNN enables joint certification. We note that DPA (Levine & Feizi, 2020) proposed to use a hash function to assign training examples into partitions, which is different from our use of hash function. In particular, we use a hash function to rank training examples. Moreover, both DPA and our work rank the labels to break ties when comparing them with respect to their votes.

A line of works (Wilson, 1972; Guyon et al., 1996; Peri et al., 2019; Bahri et al., 2020) leveraged nearest neighbors to clean a training dataset. For instance, Wilson (1972) proposed to remove a training example whose label is not the same as the majority vote among the labels of its 3 nearest neighbors. Peri et al. (2019) proposed to remove a training example whose label is not the mode amongst labels of its k nearest neighbors in the feature space. Bahri et al. (2020) combined kNN with an intermediate layer of a preliminary deep neural network model to filter suspiciously-labeled training examples. Another line of works (Gao et al., 2018; Reeve & Kabán, 2019) studied the resistance of nearest neighbors to random noisy labels. For instance, Gao et al. (2018) analyzed the resistance of kNN to asymmetric label noise and introduced a Robust kNN to deal with noisy labels. Reeve & Kabán (2019) further analyzed the Robust kNN proposed by (Gao et al., 2018) in the setting with unknown asymmetric label noise. kNN and its variants have also been used to defend against adversarial examples (Wang et al., 2018; Sitawarin & Wagner, 2019a; Papernot & McDaniel, 2018; Sitawarin & Wagner, 2019b; Dubey et al., 2019; Yang et al., 2020; Cohen et al., 2020). For instance, Wang et al. (2018) analyzed the robustness of nearest neighbors to adversarial examples and proposed a more robust 1-nearest neighbor. Several works (Amsaleg et al., 2017; Wang et al., 2018; 2019; Yang et al., 2020) proposed adversarial examples to nearest neighbors, e.g., Wang et al. (2019) proposed adversarial examples against 1-nearest neighbor. These works are orthogonal to ours as we focus on analyzing the certified robustness of kNN and rNN against general data poisoning attacks.

## 6 CONCLUSION AND FUTURE WORK

In this work, we derive the certified robustness of nearest neighbor algorithms, including kNN and rNN, against data poisoning attacks. Moreover, we derive a better lower bound of certified accuracy for rNN via jointly certifying multiple testing examples. Our evaluation results show that 1) both kNN and rNN outperform state-of-the-art certified defenses against data poisoning attacks, and 2) joint certification outperforms individual certification. Interesting future work includes 1) extending joint certification to other learning algorithms, 2) improving joint certification via new grouping methods, and 3) improving certified accuracy of kNN and rNN via new distance metrics.

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

## A    PROOF OF THEOREM 2

*Proof.* We have the following:

$$CA(e) = \min_{D_{tr}^* \in S(D_{tr}, e)} \frac{\sum_{(\mathbf{x}_i, y_i) \in D_{te}} \mathbb{I}(\mathcal{M}(D_{tr}^*, \mathbf{x}_i) = y_i)}{|D_{te}|} \tag{8}$$

$$\geq \frac{\sum_{(\mathbf{x}_i, y_i) \in D_{te}} \min_{D_{tr}^* \in S(D_{tr}, e)} \mathbb{I}(\mathcal{M}(D_{tr}^*, \mathbf{x}_i) = y_i)}{|D_{te}|} \tag{9}$$

$$= \frac{\sum_{(\mathbf{x}_i, y_i) \in D_{te}} \mathbb{I}(a_i = y_i) \min_{D_{tr}^* \in S(D_{tr}, e)} \mathbb{I}(\mathcal{M}(D_{tr}^*, \mathbf{x}_i) = a_i)}{|D_{te}|} \tag{10}$$

$$= \frac{\sum_{(\mathbf{x}_i, y_i) \in D_{te}} \mathbb{I}(a_i = y_i) \mathbb{I}(e \leq e_i^*)}{|D_{te}|}, \tag{11}$$

where the last step is based on applying Theorem 1 to testing input $\mathbf{x}_i$. □

## B    PROOF OF THEOREM 4

*Proof.* We have the following:

$$CA(e) = \min_{D_{tr}^* \in S(D_{tr}, e)} \frac{\sum_{(\mathbf{x}_i, y_i) \in D_{te}} \mathbb{I}(\mathcal{M}(D_{tr}^*, \mathbf{x}_i) = y_i)}{|D_{te}|} \tag{12}$$

$$= \min_{D_{tr}^* \in S(D_{tr}, e)} \frac{\sum_{j=1}^{\lambda} \sum_{(\mathbf{x}_i, y_i) \in \mathcal{U}_j} \mathbb{I}(\mathcal{M}(D_{tr}^*, \mathbf{x}_i) = y_i)}{\sum_{j=1}^{\lambda} |\mathcal{U}_j|} \tag{13}$$

$$\geq \frac{\sum_{j=1}^{\lambda} \min_{D_{tr}^* \in S(D_{tr}, e)} \sum_{(\mathbf{x}_i, y_i) \in \mathcal{U}_j} \mathbb{I}(\mathcal{M}(D_{tr}^*, \mathbf{x}_i) = y_i)}{\sum_{j=1}^{\lambda} |\mathcal{U}_j|} \tag{14}$$

$$\geq \frac{\sum_{j=1}^{\lambda} \mu_j \cdot |\mathcal{U}_j|}{\sum_{j=1}^{\lambda} |\mathcal{U}_j|}, \tag{15}$$

where we have Equation (15) from (14) based on applying Theorem 3 to group $\mathcal{U}_j$. □

