# OpenReview forum: "Certified Robustness of Nearest Neighbors against Data Poisoning Attacks"
_ICLR.cc/2021/Conference — Reject_

### Official Review · AnonReviewer2 · 2020-10-18
**Minimal Theoretical Contribution; Little empirical value; Existence of much stronger results**

**Rating:** 3
**Confidence:** 4

**Review:**

Existence of much stronger results:
I don't get why majority voting is claimed to be the "state-of-the-art" technique. If I'm understanding it correctly, the majority vote technique can only handle a number of corruption points up to O(K), K being the number of voters. Furthermore, since the voters more or less split the dataset, in order to maintain the accuracy of each voter, the number of voters can't be too large, and is usually O(1). Therefore, this majority vote approach can only handle O(1) number of corruption.

More specifically, for the case of kNN, the certified accuracy (Theorem 2, ) becomes vacuous as soon as the number of corrupted points $e$ becomes greater than $k$.

On the other hand, there are techniques developed from the robust statistics community that can be robust against an $\epsilon$-fraction of corruption points, that is, if there are a total of $N$ training points, it allows $\epsilon N$ number of points to be corrupted. For example, Sever [1], a recently developed robust supervised learning algorithm, guarantees $O(\sqrt{\epsilon})$ generalization error under $\epsilon$-fraction of arbitrary corruption. Such a guarantee is much stronger than the ones majority voting approaches are able to achieve.

Thus, I'm having trouble appreciating the contribution of this paper given the existence of much stronger results.

Relevance to the field:
While prior approaches like DPA also suffers from the same weak/trivial guarantee, they are at least meta-algorithms that allows one to plug in any base learners depending on the application. The method developed in this paper, however, only works on kNN. And let's be honest, not many modern ML applications use kNN with the slightest chance. So I don't see much empirical value nor any significant theoretical contribution.

[1] Ilias Diakonikolas, Gautam Kamath, Daniel M. Kane, Jerry Li, Jacob Steinhardt, Alistair Stewart. Sever: A Robust Meta-Algorithm for Stochastic Optimization.

---

> ### Comment · ~Jinyuan_Jia2 · 2021-02-02
> **Author Response**
>
> Thanks for the review.
>
> Comment 1: Much stronger results (Diakonikolas et al. (2019)).
>
> Response 1: Diakonikolas et al. (2019)  proposed a robust meta-algorithm which iteratively removes outliers such that the model parameters learnt on the poisoned dataset are close to those learnt on the clean dataset under certain assumptions. However, given a testing dataset, their method cannot compute the certified classification accuracy.
>
> Comment 2: Not many modern ML applications use kNN.
>
> Response 2: We can leverage self-supervised learning to improve the performance of kNN and rNN. In particular, we can use a pre-trained encoder as a feature extractor (Radford et al. (2021)) and our experimental results show that it can significantly improve the certified accuracy of kNN and rNN. For instance, when an attacker can arbitrarily poison 500 training examples on CIFAR10, the certified accuracy of kNN improves by 40.6% if we use CLIP (Radford et al., 2021), a feature extractor pre-trained via self-supervised learning, to extract features for each training or testing input.
>
> Radford et al. “Learning Transferable Visual Models From Natural Language Supervision.”. In Arxiv, 2021.

---

### Official Review · AnonReviewer1 · 2020-10-27
**Review for Certified Robustness of Nearest Neighbors against Data Poisoning Attacks**

**Rating:** 5
**Confidence:** 4

**Review:**

The paper studies robustness of k-NN and r-NN against data poisoning attacks. The main message of the paper is that k-NN and r-NN are automatically resilient against attacks. Furthermore, by grouping test examples based their predicted labels. Data points with different predictions are grouped together, and then better certification guarantee can be derived. Experimental results demonstrate that k-NN and r-NN are indeed self-robust against data poisoning attacks.

The theoretical angle proposed in this paper is interesting. However, I have a high-level question regarding the paper. Is the goal of the paper simply providing theoretical robustness guarantee of k-NN? I do not see any new defense mechanisms developed in this paper. The grouping idea is only intended to prove the theoretical results. If so, I feel like contribution of the paper is not significant enough. Besides that, how does the result in this paper compare to the following one? I hope the authors could clarify the concern.
Analyzing the robustness of nearest neighbors to adversarial examples

Secondly, I think definition (1) is not explained well. Does modification, addition and deletion counts equally as a single operation? The authors seem to point toward that modification is equivalent to one-time addition plus on-time deletion. This makes me wondering if modification counts as two operations? Furthermore, does the training data allow repetitions of any data point? For example, if the attacker simply adds the same point that exists in the clean training data multiple times? Does that count as poisoning? In definition (1), the dataset is considered a set, thus repetitions will be absorbed as a single item, which leads to D*=D. However, repetitions definitely matter for k-NN, and could be potentially exploited by the attacker. Therefore, I hope the authors could provide a more clear definition of the defining poisoning size.

In the beginning of section 3.1, a data point is certifiably correct only when the predicted label stays unchanged before and after attack, and it matches the true label. I am wondering does this requirement rule out the test examples that are originally misclassified? There are indeed cases where due to the attack, some test examples become correctly classified, while originally they are not. In this case, I am not sure how to interpret the robustness because the attacker is conversely helping the k-NN. This makes wonder if certification is a correct way to define robustness. Throughout the paper, there is no discussion regarding the ground-truth underlying data distribution, which I think is important for the definition of robustness. Ideally, a robust classifier should maintain high accuracy over the underlying data distribution even with attack. There is little reason to care too much about the certification for a particular data point, since that point appears with probability 0. I wonder if the authors can discuss this point.

The theoretical results in this paper are very interesting, but it lacks a nice and intuitive explanation before the proof. For example, why grouping can give us a better certification rate? What is the intuition behind that? The explanation is not given enough space in this paper. I only see a few sentences before Theorem 3, and that does not support my understanding a lot. In Figure 1, I sort of see why multiple data points cannot be jointly certified. In my understanding, it is because the point 0 and 2 are somewhat far from each other, thus the attacker cannot place enough poisoning points within the overlapping neighborhood of the two points such that their labels are changed simultaneously. However, if they are close enough, then the attacker can modify some neighbors of point 2 while adding new points to the overlapping area, which leads to a successful attack. Therefore, it is not easy to distill the distinction between individual certification and joint certification from this example. This example also does not help understanding of later theorems. I am wondering if the authors could design a more informative example?

In the experiments, I think it will be helpful to also draw the theoretical lower bound for the individual and joint certification, so that readers can see whether the analysis is tight or not. Apart from that, what is the data poisoning attack algorithm used to evaluate the robustness?

Overall, I believe the paper is not written clearly, and there is a huge space for improvement.

---

> ### Comment · ~Jinyuan_Jia2 · 2021-02-02
> **Author Response**
>
> Thanks for the review.
>
> Comment 1: What is the data poisoning attack algorithm used to evaluate the robustness.
>
> Response 1: Our certified accuracy holds for arbitrary data poisoning attacks once the poisoned size is no larger than e.

---

### Official Review · AnonReviewer5 · 2020-11-05
**Limited technical novelty compensated by simplicity and impressive empirical performance**

**Rating:** 4
**Confidence:** 3

**Review:**

**Summary:**
First, the paper identifies k-Nearest Neighbor (kNN) and radius Nearest Neighbor (rNN) to be naturally effective baseline certified defenses against data poisoning attack. It is easy to see that kNN and rNN are resistant to poison attacks, since to flip the prediction of a test example, one would need to insert/delete enough examples to change the majority vote. Second, the paper proposes a joint certificate that further improves certified accuracy for rNN. Specifically, it uses the fact that for any given poison removal budget, it can only decrease the vote for a single label. Even though the idea is simple, the experimental result is quite impressive significantly outperforming the previous more sophisticated certified defense methods.

**Strength:**
- The approach is easy to implement and understand
- Despite its simplicity, the approach performs significantly better compared to previous methods. This should be the new standard baseline for all certified poisoned defense papers.

**Weakness:**
- The technical novelty is not very strong, since it is obvious that kNN is naturally robust to poisoning. The proposed joint certification helps compensate for the technical deficiencies. However, I expect more ways to improve this lower bound than what is presented here. For example, another natural way to improve the joint certification is to consider how the added poison cannot influence two test examples concurrently when they are far enough apart.
- Even though joint certification can improve certified accuracy, in practice, individual certification may be more important compared to joint certification, since users of the system probably want certifications as individual queries come in.
- Since kNN/rNN are not used as frequently in practical settings, the proposed solution may not be as useful to systems that require the use of neural networks. However, I am also aware that none of the existing defenses work well enough for any practical setting.


**Recommendation**
Even though the technical novelty is limited, I recommend acceptance due to the simplicity of the approach and its impressive performance compared to previous methods. I think this paper will become a new standard baseline for future certified poisoned defense papers.

**Update**
After reading reviewer2's comment, I realize that there is literature proving much stronger results that I was unaware of. I still think these results should be used as a standard baseline for certified poisoning defense, but due to the lack of novelty, I have to downgrade my score.

---

> ### Comment · ~Jinyuan_Jia2 · 2021-02-02
> **Author Response**
>
> Thanks for the review.
>
> Comment 1: kNN and rNN are not used frequently in practical settings.
>
> Response 1: We can leverage self-supervised learning to improve the performance of kNN and rNN. In particular, we can use a pre-trained encoder as a feature extractor (Radford et al. (2021)) and our experimental results show that it can significantly improve the certified accuracy of kNN and rNN. For instance, when an attacker can arbitrarily poison 500 training examples on CIFAR10, the certified accuracy of kNN improves by 40.6% if we use CLIP (Radford et al., 2021), a feature extractor pre-trained via self-supervised learning, to extract features for each training or testing input.
>
> Radford et al. “Learning Transferable Visual Models From Natural Language Supervision.”. In Arxiv, 2021.
>
> Comment 2: Much stronger results (Diakonikolas et al., 2019).
>
> Response 2: Diakonikolas et al. (2019)  proposed a robust meta-algorithm which iteratively removes outliers such that the model parameters learnt on the poisoned dataset are close to those learnt on the clean dataset under certain assumptions. However, given a testing dataset, their method cannot provide the certified classification accuracy.

---

### Official Review · AnonReviewer3 · 2020-11-06
**Neat theoretical results; questionable for a practical application**

**Rating:** 5
**Confidence:** 2

**Review:**

This paper studies to train a certifiable robust model against data poisoning attacks using nearest neighbors. The paper studies the voting mechanism in the nearest neighbor models, and presents a relationship between the poisoning instances and the difference between the majority votes and the second majority votes. Such a relationship will result in a guarantee on the lower bound of a training model's accuracy, which is referred to as Certified Accuracy (CA). The theoretical results are neat. The experiments are conducted on MNIST and CIFAR, and results show better CA than previous approaches of DPA and Bagging.

My main concern is the limitation of the applicable machine learning models, which seems restricted to only kNN and rNN models --- they may not yield the best performance on most interesting tasks. For example, from Fig 2 and 3, we can see that even when poisoning size e=0, the accuracy (which should be identical to CA) is far below the SOTA on the corresponding MNIST and CIFAR tasks.

Also, the lower bound is established with respect to s_a - s_b - e <= k-e. Therefore, to be able to handle larger poisoning size e, one has to employ a larger k (in the kNN case) or larger r (in the rNN case). Such choices of hyperparameter typically hinder the accuracy on the clean dataset. It is not clear how such a restriction can be mitigated in a practical setup.

Due to the above concern, I'm borderline on work.

---

> ### Comment · ~Jinyuan_Jia2 · 2021-02-02
> **Author Response**
>
> Thanks for the review.
>
> Comment 1: kNN and rNN may not yield the best performance on most interesting tasks.
>
> Response 1: We can leverage self-supervised learning to improve the performance of kNN and rNN. In particular, we can use a pre-trained encoder as a feature extractor (Radford et al. (2021)) and our experimental results show that it can significantly improve the certified accuracy of kNN and rNN. For instance, when an attacker can arbitrarily poison 500 training examples on CIFAR10, the certified accuracy of kNN improves by 40.6% if we use CLIP (Radford et al., 2021), a feature extractor pre-trained via self-supervised learning, to extract features for each training or testing input.
>
> Radford et al. “Learning Transferable Visual Models From Natural Language Supervision.”. In Arxiv, 2021.

---

### Official Review · AnonReviewer4 · 2020-11-07
**Theorem 3 is problematic**

**Rating:** 4
**Confidence:** 4

**Review:**

The paper studies the robustness of kNN and rNN to poisoning attacks. They try to achieve certified bounds on the robustness of rNN and kNN algorithms, against poisoning attacks. They claim to achieve certified upper bounds on the effect of poisoning attacks on overall accuracy. The techniques used are very simple. Their first theorem which is about the robustness of individual test examples states that if the number of poison points are smaller than half the distance between the number of neighbors with label that has highest vote and the number neighbors with label that has second highest vote, then poisoning is not effective. Then they try to extend this result to overall accuracy of scheme. Their Theorem 3 tries to achieve such a bound but I think this theorem is not correct (I mentioned the issue with this Theorem in the comments bellow). Then they have some experiments that uses this Theorem to achieve certified bounds for the case of  MNIST and CIFAR10.

Their experimental results show that they can achieve better certified bounds compared to some previous certified defense papers. But their results are extremely dependent on the correctness of Theorem 3 and it is necessary for authors to rewrite the proof and theorem itself.

Even if the authors fix their theorem, I still don't find the theoretical contribution of this paper significant. But their experiments and certified bounds could be interesting enough for paper to be accepted.


Comments to Authors:

- I think there is a problem in the  proof of Theorem 3. It is mentioned that a_i's are different. But b_is are not necessarily different and they could all be the same label. Then the attacker can potentially flip all of them to b_i.

-In page 4, it is mentioned that for breaking the ties in kNN we can use ranking of training examples. How would that exactly work?

-The claim on being the first certified poisoning defense on overall accuracy is not correct. The work of Steinhardt et al (2017) also studies provable defenses for overall accuracy. It is crucial for this paper to compare their result with Steinhart et al (2017).

page 1: worse-case -> worst-case
page 2:

page 3: The formal definition o S(D_tr,e) is inconsistent with what is described in text.

Page 4: The notation is not very clear. s_l is the number of votes for class l and for instance x. It is better to use something like s_l(x). Similarly, a_i and b_i are better to be a_i(x) and b_i(x).

---

> ### Comment · ~Jinyuan_Jia2 · 2021-02-02
> **Author Response**
>
> Thanks for the review.
>
> Comment 1: The problem in the proof of Theorem 3.
>
> Response 1: Sorry for the confusion. In our proof of Theorem 3, we don’t require that the b_i’s are different, i.e., all b_i’s can be the same. In other words, our bound still holds if an attacker flips all of them to the same b_i’s.
>
> Comment 2: Comparing with Steinhard et al (2017).
>
> Response 2: Steinhard et al (2017) derived an upper bound of the loss function for data poisoning attacks when the model is learnt using examples in a feasible set. However, their method cannot provide the certified classification accuracy for a given testing dataset.

---

### Decision · Program_Chairs · 2021-01-07
**Final Decision**

**Decision:**

Reject

**Comment:**

Some reviewers expressed concerns on soundness of the theory in the paper. Specifically, theorem 3 does not seem to be correct.  There are other concerns such as the significance of the theoretical contributions, little empirical value and existence of much stronger results. Unfortunately the authors did not provide responses to the concerns raised by the reviewers.

---

> ### Comment · ~Jinyuan_Jia2 · 2021-02-02
> **Author Response**
>
> Thanks for the comments. We have clarified those major concerns in the response to each reviewer.